# An Attentive Approach for Building Partial Reasoning Agents from Pixels

**Safa Alver**                                                                                  *safa.alver@mail.mcgill.ca*
*Mila, McGill University*

**Doina Precup**                                                                                *dprecup@cs.mcgill.ca*
*Mila, McGill University and Google DeepMind*

**Reviewed on OpenReview:** *https://openreview.net/forum?id=S3FUKFMRw8*

## Abstract

We study the problem of building reasoning agents that are able to generalize in an effective manner. Towards this goal, we propose an end-to-end approach for building model-based reinforcement learning agents that dynamically focus their reasoning to the relevant aspects of the environment: after automatically identifying the distinct aspects of the environment, these agents dynamically filter out the relevant ones and then pass them to their simulator to perform partial reasoning. Unlike existing approaches, our approach works with pixel-based inputs and it allows for interpreting the focal points of the agent. Our quantitative analyses show that the proposed approach allows for effective generalization in high-dimensional domains with raw observational inputs. We also perform ablation analyses to validate our design choices. Finally, we demonstrate through qualitative analyses that our approach actually allows for building agents that focus their reasoning on the relevant aspects of the environment.

## 1 Introduction

The goal in model-based reinforcement learning (RL) is to build reasoning agents that maximize long-term cumulative reward through trial and error interaction with the environment. In the last decade, the employment of deep neural networks as function approximators have allowed model-based RL agents to achieve significant successes across a wide range of areas, ranging from challenging board and video games (Silver et al., 2017b; 2018; Ha & Schmidhuber, 2018; Hafner et al., 2020; 2021; 2023; Schrittwieser et al., 2020) to discovering efficient algorithms (Fawzi et al., 2022; Mandhane et al., 2022; Mankowitz et al., 2023). However, despite these successes, these agents are typically designed to improve sample efficiency in regular RL settings and thus lack the inductive biases that are necessary for achieving effective performance in generalization settings.

One of the promising lines of research for tackling this shortcoming considers the use of *partial models* in the reasoning process of the agent (see e.g. Talvitie & Singh, 2008; Khetarpal et al., 2020; 2021; Zhao et al., 2021; Alver & Precup, 2023). Inspired by studies in cognitive science, these approaches aim for building agents that mimic the reasoning process in humans, in which planning is hypothesized to be performed over a few abstract aspects of the environment (Bengio, 2017; Goyal & Bengio, 2022): e.g. when planning a route from point A to point B humans typically focus their reasoning to the points in between and discard the rest. More specifically, rather than building plain reasoning agents that plan over every aspect of the environment, these approaches advocate for building partial reasoning agents that focus the planning on the relevant aspects of the environment. Among these studies, the study of Zhao et al. (2021) has empirically demonstrated that this type of planning allows for effective generalization in scenarios where the details of the environment keep changing but the overall task remains the same. However, this study has two main shortcomings: (i) it fails in providing an approach for building partial reasoning agents that works with pixel-based inputs, i.e. an

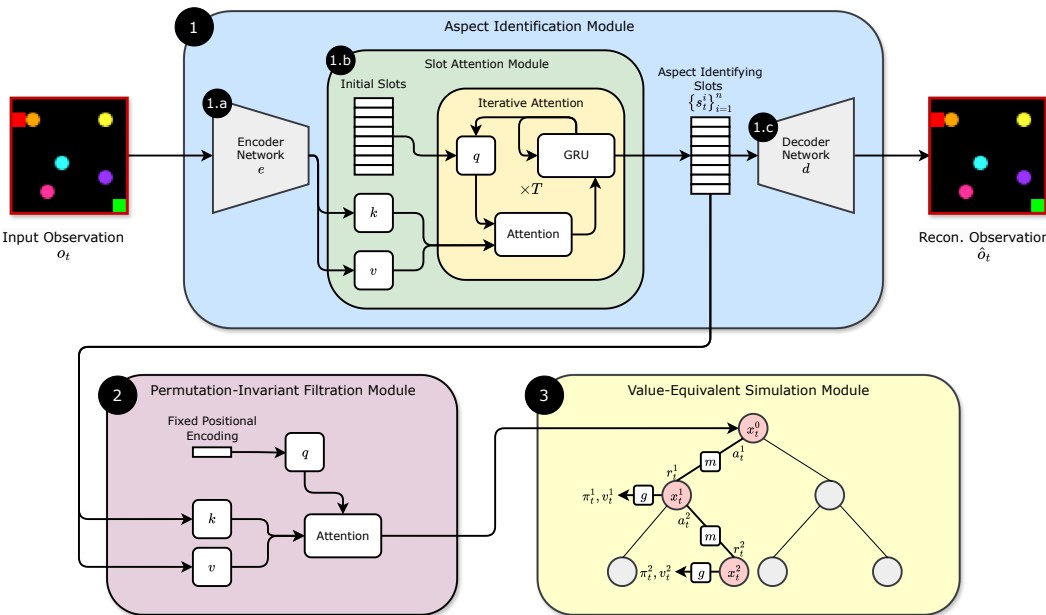

Figure 1: Our overall architecture for building partial reasoning agents. This architecture consists of three main modules: (**Box 1**) the aspect identification module, (**Box 2**) the permutation-invariant filtration module, and (**Box 3**) the value-equivalent simulation module. The details of these modules are provided in the main text. The $q$, $k$ and $v$ blocks represent a linear mapping to obtain the query, key and value matrices, respectively, and the attention blocks represent a soft attention operation. The GRU block represents a GRU cell. Lastly, the black arrows indicate the direction of information flow. More details on this architecture can be found in App. A.

approach that works beyond low-dimensional environments in which the distinct aspects of the environment is hand-provided via symbolic inputs, and (ii) it lacks detailed qualitative analyses demonstrating that their proposed approach actually allows for building agents that focus their reasoning on the relevant aspects of the environment.

In this paper, we overcome these shortcomings by proposing an approach for building partial reasoning agents that works with pixel-based inputs and allows for interpreting the focal points of the agent. Unlike prior work, our end-to-end approach allows for building reasoning agents that automatically identify the distinct aspects of the environment and then dynamically attend to the relevant ones, which makes it possible for them to work on high-dimensional domains with raw observational inputs. Importantly, the built agents also allow for further inspection on what they actually incorporate into their model throughout the course of interaction with the environment. After laying out its details (Sec. 3), we demonstrate through quantitative analyses that our approach allows for effective generalization in high-dimensional domains (Sec. 4.3). In addition to the quantitative analyses, we also perform ablation analyses to validate our design choices (Sec. 4.4). Finally, we perform several qualitative analyses with our approach and show that it actually allows for building agents that dynamically focus their reasoning on the relevant aspects of the environment (Sec. 4.5). We hope that our study will bring the RL community a step closer to building scalable and interpretable reasoning agents that are able to effectively generalize to novel situations throughout their interaction with the environment.

**Key Contributions.** The key contributions of this study are as follows: (i) We propose an approach for building partial reasoning agents that works with pixel-based inputs and allows for interpreting the focal points of the agent. (ii) We demonstrate, through quantitative analyses, that our end-to-end approach allows for effective generalization in high-dimensional domains with raw observational inputs. (iii) We demonstrate, through qualitative analyses, that our approach allows for building agents that actually focus their reasoning on the relevant aspects of the environment.

## 2   Background

**Reinforcement Learning.** In RL (Sutton & Barto, 2018), an agent interacts with its environment through a sequence of actions to maximize its long-term cumulative reward. The environment is usually modeled as a Markov decision process (MDP) $M \equiv (\mathcal{S}, \mathcal{A}, P, R, d_0, \gamma)$ where $\mathcal{S}$ and $\mathcal{A}$ are the (finite) set of states and actions, $P : \mathcal{S} \times \mathcal{A} \rightarrow \mathrm{Dist}(\mathcal{S})$ is the transition distribution, $R : \mathcal{S} \times \mathcal{A} \times \mathcal{S} \rightarrow \mathbb{R}$ is the reward function, $d_0 : \mathcal{S} \rightarrow \mathrm{Dist}(\mathcal{S})$ is the initial state distribution and $\gamma \in [0, 1)$ is the discount factor. At each time step $t$, after taking an action $a_t \in \mathcal{A}$, the environment's state transitions from $s_t \in \mathcal{S}$ to $s_{t+1} \in \mathcal{S}$; and the agent receives an observation $o_{t+1} \in \mathcal{O}$ (reflecting $s_{t+1}$) and an immediate reward $r_t$. The goal of the agent is to learn a policy $\pi : \mathcal{O} \rightarrow \mathrm{Dist}(\mathcal{A})$ that maximizes $E_{\pi,P}[\sum_{t=0}^{\infty} \gamma^t R(S_t, A_t, S_{t+1}) | S_0 = s_0 \sim d_0]$, where $E_{\pi,P}[\cdot]$ denotes the expectation over trajectories induced by $\pi$ and $P$.

**Model-Based RL.** One way of achieving this goal is through model-based RL (Moerland et al., 2023) in which there are two alternating phases: (i) the learning and (ii) planning phases. In the learning phase, the gathered experience is mainly used in learning a model $m$, and in the planning phase, the learned model $m$ is then used for simulating experience either to be used alongside real experience in improving the value predictions or just to be used in selecting actions at decision time (Alver & Precup, 2024).

**The Attention Mechanism.** In its general form, the soft attention mechanism (Vaswani et al., 2017) is described as $y = \sigma(QK^{\top})V$ where $Q \in \mathbb{R}^{m \times d_q}$, $K \in \mathbb{R}^{n \times d_q}$ and $V \in \mathbb{R}^{n \times d_v}$ are the query, key and value matrices, respectively, and $\sigma(\cdot)$ is the softmax function. Here, $m$ and $n$ denote the number of query and key / value vectors, respectively; and, $d_q$ and $d_v$ denote the dimensionality of the query / key and value vectors. More explicitly, the attention output $y$ is computed in three steps: (i) first, each of the $m$ $d_q$-dimensional query vectors are dotted with a database of $n$ $d_q$-dimensional key vectors to compute the score values, (ii) then these values are passed through a softmax function $\sigma(\cdot)$, along the axis of the key vectors, to compute the attention weights, and (iii) finally, the output $y$ is computed by taking a weighted sum of the $n$ $d_v$-dimensional value vectors with the weights.

**Self-Attention and Permutation-Invariance.** A particular form of attention that is of interest in this study is self-attention. In self-attention, the $Q$, $K$, $V$ matrices are a function of the input $x$, e.g., $Q = q(x)$, $K = k(x)$ and $V = v(x)$ where $q(\cdot)$, $k(\cdot)$ and $v(\cdot)$ are linear mappings. By default, the self-attention operation is not permutation-invariant, i.e., permuting the input also results in a permutation in the output. However, a clever way to enable permutation-invariance in self-attention is to make $Q$ a function of a set of fixed positional embeddings, rather than the input $x$ (see Lee et al., 2019; Tang & Ha, 2021).

## 3   An Attentive Approach for Building Partial Reasoning Agents from Pixels

In this section, we present our approach for building partial reasoning agents that works with pixel-based inputs and allows for interpreting the focal points of the agent. Our approach consists of an end-to-end architecture and an accompanying training procedure. The architecture, depicted in Fig. 1, consists of three separate modules: (i) the aspect identification module, (ii) the permutation-invariant filtration module and (iii) the value-equivalent simulation module. The details of these modules and the procedure for collectively training them are presented in the following sections.

### 3.1   Aspect Identification Module

At a high level, at each time step $t$ in the environment, the aspect identification module (see Box 1 in Fig. 1) takes in the raw observational input $o_t$ and binds its distinct aspects into a set of $n$ distinct slots $\{s_t^i\}_{i=1}^n$. The main function of this module is to automatically identify the distinct aspects of the environment, e.g. the distinct objects or the distinct regions in the observational input. Internally, this module resembles an autoencoder architecture (Goodfellow et al., 2016) and it consists of (i) an encoder network, (ii) a slot attention module[1] and (iii) a decoder network.

---

[1] We prefer slot attention over other techniques as it allows for better generalization (Locatello et al., 2020). We also note that even though Locatello et al. (2020) introduced slot attention and used it for the purposes of object discovery and set prediction in the context of computer vision, in this study, we are using slot attention for the purpose of building partial reasoning agents.

The encoder network $e$ (see Box 1.a in Fig. 1) takes as input the raw observation $o_t$ and encodes it into vectorized representations $f_t$ that are augmented with positional embeddings, i.e. $f_t = e(o_t)$. It follows a simple structure: a convolutional backbone followed by a positional embedding layer.

The vectorized representations $f_t$ are then passed through the slot attention module (Locatello et al., 2020) to obtain a set of $n$ aspect identifying slots $\{s_t^i\}_{i=1}^n$, which form a set with permutation symmetry. Each slot in the set can, for example, describe an object or a certain region in the observational input. This module uses an iterative attention mechanism to produce the set of slots. More specifically, slots are initialized at random and thereafter refined for $T$ iterations with a Gated Recurrent Unit (GRU, Cho et al., 2014) to bind to particular grouping of the vectorized representations $f_t$ (here each iteration refers to a single recurrence of the GRU). At each iteration, slots "compete" for the reconstructing parts of the input via a soft attention mechanism (Vaswani et al., 2017). The internal details of this module are depicted in Box 1.b of Fig. 1. For more details, we refer the reader to the study of Locatello et al. (2020).

Finally, the aspect identifying slots $\{s_t^i\}_{i=1}^n$ are then passed through the decoder network $d$ (see Box 1.c in Fig. 1) to obtain the observation reconstruction $\hat{o}_t$, i.e. $\hat{o}_t = d(\{s_t^i\}_{i=1}^n)$. The decoder network is a spatial broadcast decoder (Watters et al., 2019), as in (Greff et al., 2019): each of the aspect identifying slots are broadcasted into a 2-dimensional grid and augmented with positional embeddings. Each grid is decoded using a convolutional network to produce an output of size $W \times H \times 4$, where $W$ and $H$ are the height and width of the image, respectively. The output channels encode the RGB color channels and an unnormalized alpha mask. The alpha masks are then normalized using a softmax across the slots and used as mixture weights to combine the individual reconstructions into a single RGB image, which we refer to as $\hat{o}_t$.

For more on the details of aspect identification module, we refer the reader to App. A.1.

### 3.2 Permutation-Invariant Filtration Module

The permutation-invariant (PI) filtration module (see Box 2 in Fig. 1) takes as input the aspect identifying slots $\{s_t^i\}_{i=1}^n$, i.e. the output of the slot attention module, and performs a "soft" filtration operation over these slots to distill out the relevant information. The main purpose of this module is to filter out the relevant information from the slots that are likely to be useful in decision-making, which is later to be used for partial reasoning. Here, it is important to note that the filtration is dynamic, i.e. throughout the agent-environment interaction, the PI filtration module can filter out different slots at different time steps. Internally, the slots are weighted by a PI soft attention mechanism (Lee et al., 2019). Here, the use of a PI attention mechanism is critical, as the aspect identifying slots form a set with permutation symmetry, i.e. the order of them can change at each time step. More on the details of this module can be found in App. A.2.

It is also important to note that the PI filtration module has some analogues to the "consciousness-inspired bottleneck" that was proposed in Zhao et al. (2021). However, there is an important difference: rather than using a top-k semi-hard attention mechanism (Ke et al., 2018), we use a relatively straightforward soft attention mechanism (Vaswani et al., 2017). This difference makes our approach much simpler and it also mitigates the burden of tuning for the optimal k value. Experiments in the following sections also demonstrate that the use of top-k semi-hard attention does not bring any performance advantages.

### 3.3 Value-Equivalent Simulation Module

The value-equivalent simulation module (see Box 3 in Fig. 1) takes the filtered aspect identifying slots as input and uses them as the root state $x_t^0$ for performing value-equivalent simulation (Grimm et al., 2020; 2021; Schrittwieser et al., 2020). The main function of this module is to perform partial reasoning, i.e. to perform planning with the relevant aspects of the environment. More specifically, at each time step $t$, for each of the hypothetical $k = 1, \ldots, K$ steps, the simulator takes in the root state $x_t^0$ and predicts three future quantities: (i) the policy $p_t^k \approx \pi(a_{t+k+1}|o_1, \ldots, o_t, a_{t+1}, \ldots, a_{t+k})$, (ii) the value function $v_t^k \approx E[u_{t+k+1} + u_{t+k+2} + \ldots|o_1, \ldots, o_t, a_{t+1}, \ldots, a_{t+k}]$, and (iii) the immediate reward $r_t^k \approx u_{t+k}$, where $u$ is the true reward, $\pi$ is the policy used to select real actions, and $\gamma$ is the discount factor. Internally, the simulator is represented by a combination of a dynamics and prediction network. The dynamics network $m$ recurrently predicts, at each hypothetical step $k$, an internal state $x_t^k$ and an immediate reward $r_t^k$, i.e.

$x_t^k, r_t^k = m(x_t^{k-1}, a_t^k)$. The policy and value function are then computed from the internal state $x_t^k$ by a prediction network $g$, i.e. $p_t^k, v_t^k = g(x_t^k)$. We refer the reader to App. A.3 for more details on this module.

Given such a simulator, it is possible to do planning over the internal state space and reward induced by the dynamics network. Specifically, in this module, we use the Monte-Carlo tree search (MCTS, Coulom, 2006) algorithm to perform the planning. At each time step $t$, this algorithm outputs a recommended policy $\pi_t$ from which an action $a_{t+1} \sim \pi_t$ is then selected.

Finally, it is important to note that, in this module, we have preferred value-equivalent simulation (see e.g. Tamar et al., 2016; Oh et al., 2017; Silver et al., 2017a; Schrittwieser et al., 2020) over other reconstruction-based simulation methods (see e.g. Watter et al., 2015; Wahlström et al., 2015; Kaiser et al., 2019; Hafner et al., 2020; 2021; 2023) as former simulation methods directly optimize for learning representations that are useful for achieving the task of interest, which also aligns with the main objective of this study.

### 3.4 Training Procedure

Using batches of data sampled from the replay buffer, the overall architecture is jointly trained, in an end-to-end manner, with a loss function that is a weighted sum of the reconstruction loss $\mathcal{L}_{\text{recon}}$ and the simulation loss $\mathcal{L}_{\text{sim}}$:

$$\mathcal{L}_{\text{total}} = \alpha \mathcal{L}_{\text{recon}}(o_t, \hat{o}_t) + \beta \underbrace{\left( \sum_{k=0}^{K} \mathcal{L}_p(\pi_{t+k}, p_t^k) + \mathcal{L}_v(z_{t+k}, v_t^k) + \mathcal{L}_r(u_{t+k}, r_t^k) \right)}_{\mathcal{L}_{\text{sim}}}, \qquad (1)$$

where $\mathcal{L}_p$, $\mathcal{L}_v$, and $\mathcal{L}_r$ are the loss functions for the policy, value function and reward, respectively, constituting the simulation loss $\mathcal{L}_{\text{sim}}$. While $\mathcal{L}_{\text{recon}}$ optimizes for minimizing the error between input observation $o_t$ and the reconstructed observation $\hat{o}_t$, $\mathcal{L}_p$, $\mathcal{L}_v$ and $\mathcal{L}_r$ optimize for minimizing (i) the error between the search policy $\pi_{t+k}$ and predicted policy $p_t^k$, (ii) the error between the improved value target $z_{t+k} = u_{t+1} + \gamma u_{t+2} + \cdots + \gamma^{n-1} u_{t+n} + \gamma^n v_{t+n}$ and the predicted value $v_t^k$, and (iii) the error between the observed reward $u_{t+k}$ and predicted reward $r_t^k$, respectively. In $\mathcal{L}_v$, the improved value targets $z_{t+k}$ are generated by collecting trajectories in the environment, and in $\mathcal{L}_p$, the improved policy targets $\pi_{t+k}$ are generated by performing MCTS. More details on individual losses and the weighting between them can be found in App. B.

## 4 Experimental Results

### 4.1 Environment Descriptions and Evaluation Setting

**Environment Descriptions.** We perform our experiments on the (i) MiniGrid (Chevalier-Boisvert et al., 2018) and (ii) Procgen domains (Cobbe et al., 2020). We choose these domains as the former one allows for designing controlled experiments and the latter is helpful in demonstrating the scalability of our approach to more complex domains. Here, it is important to note that in these domains the agent only receives raw observational inputs, i.e. 64x64 RGB images, and not symbolic inputs as in Zhao et al. (2021).[2]

In the MiniGrid domain, we consider two customized environments: (i) DynamicObstacles-Pass and (ii) DynamicObstacles-Collect (abbreviated as **DynObs-Pass** and **DynObs-Collect**, respectively, see the top row of Fig. 2). In both of these environments, the agent (red square) is randomly spawned at the leftmost column of the grid and has to pass through a region with randomly oscillating obstacles to reach the goal (green square), which is randomly placed at the rightmost column of the grid. At each time step the agent is able to move to one of its

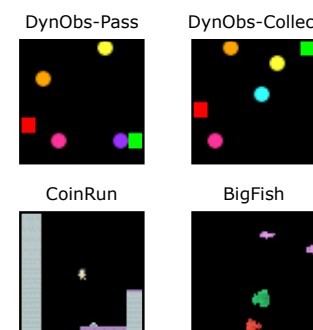

Figure 2: Sample frames from the (**top row**) DynObs-Pass, DynObs-Collect, (**bottom row**) CoinRun and BigFish environments.

---

[2]Zhao et al. (2021) perform experiments on a customized MiniGrid environment with symbolic inputs. The symbolic input is in the form of a triple $(x, y, idx)$, where $x$ and $y$ are the coordinates on the grid, and $idx$ is the id of the object.

neighboring cells. After every episode, a new game level is generated by randomly sampling obstacles among the orange, cyan, yellow, pink and purple obstacles and then by randomly placing them inside the region between the agent and the goal. In DynObs-Pass, the agent has to complete the level by dodging the existing deadly orange, cyan, pink and purple obstacles (colliding with the existing yellow obstacle has no effect). In DynObs-Collect, the agent has to complete the level by collecting the existing yellow obstacle and dodging the existing deadly orange, cyan, pink and purple ones. In both of these environments, if the agent successfully completes the level, it receives a reward of +1 and the episode terminates; otherwise the episode terminates with a total reward of 0.

In the Procgen domain, we consider two commonly-used environments: (i) **CoinRun** and (ii) **BigFish** (see the bottom row of Fig. 2). In CoinRun, the agent (an astronaut) must collect the coin that is at the far right of the level. In BigFish, the agent (a green fish) starts as a mid-sized fish and has to grow by eating the fish that are smaller than itself. In both of these environments, a new game level is procedurally generated after every episode. For more details on these environments, we refer the reader to the study of Cobbe et al. (2020).

**Evaluation Setting.** Following Zhao et al. (2021), we consider the generalization setting, in which the agent's performance is evaluated by how it performs on a set of test levels (test set) as it gets trained on a set of training levels (training set) of the same environment. Importantly, in this setting, the agent does not get to interact with the test levels, i.e. evaluation is performed in a zero-shot manner.

## 4.2 Agents

In our experiments, we compare the following agents:

**Partial Reasoning (PR) Agent.** A partial reasoning agent that was built by our proposed approach in Sec. 3.

**MuZero (MZ) Agent.** A MuZero (Schrittwieser et al., 2020) agent. Unlike the PR agent, the MZ agent performs regular reasoning. The main difference between the MZ agent and the PR agent is in the procedure for constructing the root state $x_t^0$: while the PR agent constructs it by the use of an aspect identification and PI filtration module (Sec. 3), the MZ agent constructs it by the use of a plain convolutional and residual architecture (He et al., 2016). We use this agent as a baseline to demonstrate the impact of partial reasoning in achieving better generalization.

**Model-Free (MF) Agent.** A version of the MZ agent that does not perform any planning. This agent takes actions by skipping the search process and just following its value predictions (which makes it a model-free agent). We use this agent as a baseline to demonstrate the impact of planning in achieving better generalization.

**PR-NoPIF Agent.** A version of the PR agent with no PI filtration module.

**PR-SH Agent.** A version of the PR agent in which the PI filtration module is implemented with top-k semi-hard attention. The purposes of this agent and the PR-NoPIF agent will be unfolded in the ablation analyses section (Sec. 4.4).

We note that the PR agent and the baseline agents share architectures as much as possible to ensure fair comparisons. More details on all of the agents can be found in App. C.

## 4.3 Quantitative Analyses

**Our approach allows for effective generalization in high-dimensional domains.** The study of Zhao et al. (2021) demonstrated that partial reasoning can allow for effective generalization over regular reasoning. It also demonstrated that planning plays an important role for better generalization. However, these demonstrations were mainly performed on low-dimensional gridworlds with symbolic inputs. One of the main goals of our approach is to generalize these successes to high-dimensional domains. Towards this goal, we compare the generalization performances of the PR, MZ and MF agents across two image-based MiniGrid and Procgen domains (Sec. 4.1). Performance curves in Fig. 3 consistently show that, the PR agent displays a better generalization performance than the MZ agent, both (i) highlighting the role of partial reasoning in achieving effective generalization and (ii) corroborating the findings of Zhao et al. (2021). We can also see

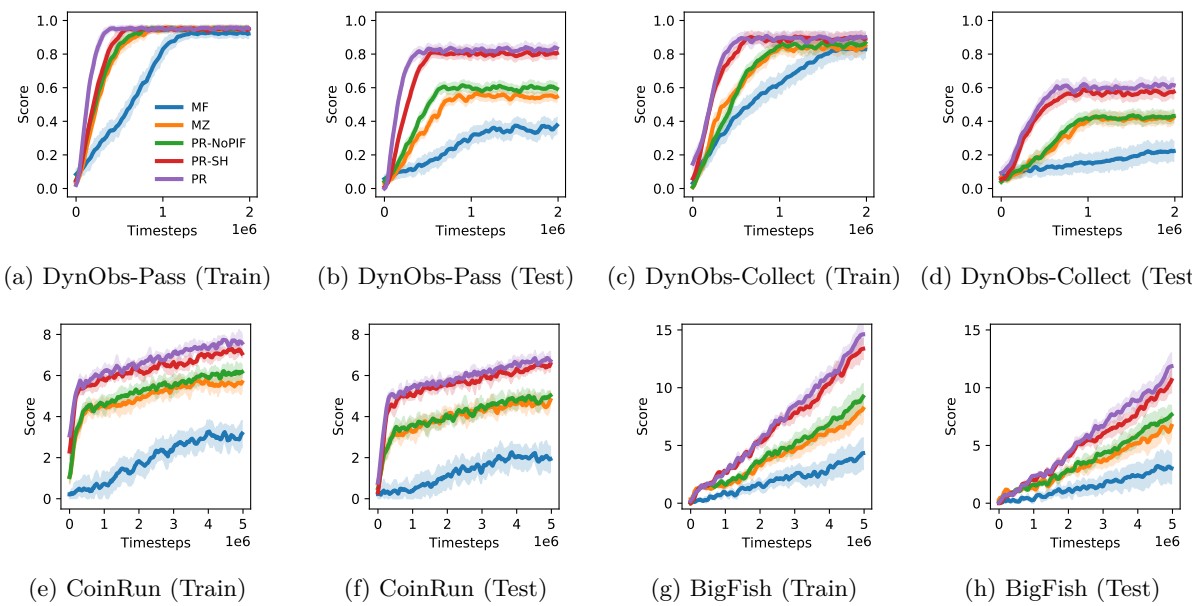

Figure 3: The generalization performances of the PR and baseline agents on the (**a**, **b**) DynObs-Pass, (**c**, **d**) DynObs-Collect, (**e**, **f**) CoinRun and (**g**, **h**) BigFish environments. Here, the plots are obtained by evaluating the agent on a set of test levels (referred to as Test) as it gets trained on a set of training levels (referred to as Train) of the same environment. In MiniGrid and Procgen experiments, the training sets consist of 16 and 500 randomly-sampled game levels, respectively. In both experiments, the test sets consist of all of the possible game levels. The means and the confidence intervals are computed over 15 independent runs.

that the MF agent underperforms the PR and MZ agents, which validates the role of planning in achieving better generalization.

## 4.4 Ablation Analyses

**The PI filtration module plays a critical role in the performance of our approach.** One of the crucial components of our proposed agent architecture is the PI filtration module (Sec. 3.2). This module is particularly important as it acts as a "soft" filter to what the agent would be modeling. To quantify its importance, we compare the PR agent with the PR-NoPIF agent (Sec. 4.2). Recall that in the PR-NoPIF agent the aspect identifying slots are directly passed, without any filtration, to the value-equivalent simulation module. Results in Fig. 3 show that the removal of the PI filtration module significantly deteriorates the generalization performance of the PR agent, making it on par with the MZ agent. These results indicate that the PI filtration module indeed plays a critical role in the performance of our approach.

**The use of top-k semi-hard attention in the PI filtration module does not bring any performance advantages.** As expressed previously, in this study, we have implemented the PR agent's PI filtration module with soft attention, in which the attention weights can take any value between zero and one. However, another way to implement this module could have been with top-k semi-hard attention, which was used in the "consciousness-inspired bottleneck" of Zhao et al. (2021) (Sec. 3.2). In top-k semi-hard attention, only the top-k attention weights are kept and all others are set to zero. If used in the PI filtration module, this type of attention would allow for a hard filtration of only the k aspect identifying slots with the largest attention weights. To investigate the impact of this type of attention, we compare the PR agent with the PR-SH agent (Sec. 4.2). The comparison results, shown in Fig. 3, indicate that the use of top-k semi-hard attention does not bring any advantages in the generalization performance of the PR agent; in fact, the use of it slightly decreases the performance. These results suggest that the use of the more simple soft attention mechanism can be a better choice in the implementation of the PI filtration module.

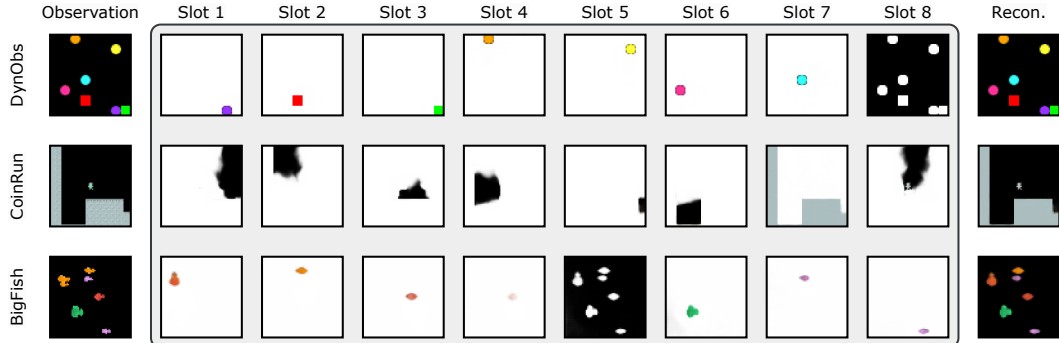

Figure 4: (**First column**) The observational inputs to the aspect identification module across different environments. (**Middle columns**) The corresponding masked reconstructions of each of the aspect identifying slots. The visualization are obtained by multiplying the alpha masks and reconstructions of each slot, which are both obtained by passing the slots through the decoder network. We note that while the aspect identifying slots bind to different objects in the DynObs-Pass, DynObs-Collect and BigFish environments, they bind to different regions in the CoinRun environment. (**Last column**) The corresponding reconstructions of the aspect identification module.

### 4.5 Qualitative Analyses

The study of Zhao et al. (2021) claimed the learning of partial reasoning agents, but lacked in providing detailed qualitative analyses that demonstrates that the learned agents actually focus their reasoning on the relevant aspects of the environment. In this section, we perform qualitative analyses with the PR agent and demonstrate that our approach actually allows for building reasoning agents that focus on the relevant aspects of the environment.

**Our approach allows for building reasoning agents that automatically identify the distinct aspects of the environment.** Before the dynamic attention illustrations, we first start by presenting visualizations to demonstrate that the PR agent is actually able to automatically identify the distinct aspects of the environment. Recall from Sec. 3.1 that the automatic identification of the distinct aspects falls under the role of the aspect identification module. For this purpose, in Fig. 4 we visualize the masked reconstructions of each of the individual aspect identifying slots across different environments. Inspecting this figure, we see that the slots indeed bind to the distinct aspects of the environment, corresponding either to different objects or to different regions in the observational input. This demonstrates that our proposed approach is indeed successful in automatically identifying the distinct aspects of the environment.

**Our approach allows for building reasoning agents that dynamically attend to the relevant aspects of the environment.** We now present illustrations to demonstrate that the PR agent is actually able to dynamically attend to the relevant aspects of the environment. Recall that the dynamic attention to the relevant aspects falls under the role of the PI filtration module (Sec. 3.2). To see if the PI filtration module is actually successful in this task, in Fig. 5, we present the attention maps of the PR agent throughout its course of interaction with different environments. Examining this figure, we can see that the PI filtration module indeed dynamically attends to the relevant aspects of different environments (see the caption of Fig. 5 for the interpretations), which demonstrates that our proposed approach indeed allows for building reasoning agents that dynamically focus on the relevant aspects of the environment.

## 5 Related Work

**Partial Reasoning Agents.** The use of partial models for building partial reasoning agents has previously been investigated in the model-based RL literature (Talvitie & Singh, 2008; Khetarpal et al., 2020; 2021; Zhao et al., 2021; Alver & Precup, 2023; Alver et al., 2024). Among these studies, the study of Zhao et al. (2021) is closest to our work in that it proposes an approach for building partial reasoning agents and

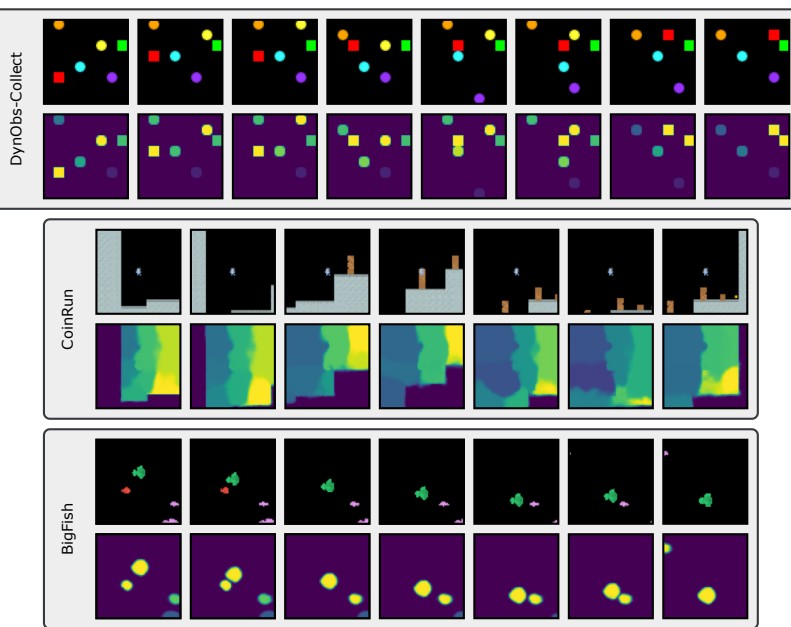

Figure 5: Example illustrations of how the attention maps of the PR agent evolve as it interacts with different environments. The attention maps are obtained by overlapping the weighted alpha masks of the aspect identifying slots, which are obtained by weighting the alpha masks with the attention weights from the PI filtration module. In the attention maps, the yellow and dark blue regions indicate the aspects with the highest and lowest attention, respectively. (**DynObs-Collect**) In the DynObs-Collect illustrations, we can see that the agent initially focuses on the yellow obstacle, while also taking into account the deadly obstacles around it, and after the collection of this obstacle it completely focuses on the goal. (**CoinRun**) In CoinRun illustrations, we can see that the agent constantly focuses on where it will land at the end of its jump while moving to the far right of the level. (**BigFish**) In BigFish illustrations, we can see that the agent initially focuses on the red fish next to it, and then after eating this fish it focuses on the pink fish in front of it.

demonstrates through quantitative analyses that their approach allows for effective generalization. However, our work differs in that we propose an approach in which the agent automatically identifies, in an end-to-end manner, the distinct aspects of the environment and then dynamically attends to the relevant ones, which makes it possible to work on domains beyond low-dimensional gridworlds in which the distinct aspects of the environment is hand-provided via symbolic inputs, i.e. high-dimensional domains with raw observational inputs. Another limitation of Zhao et al. (2021) is that they lack detailed analyses demonstrating that their proposed approach actually allows for building agents that focus on the relevant aspects of the environment. Our work overcomes this by presenting qualitative analyses in which the built partial reasoning agents dynamically attend to the relevant aspects of the environment.

Our work is also related to the study of Alver & Precup (2023) in that our partial reasoning agent is also built towards achieving effective generalization. However, rather than proposing an approach for building black-box agents that only act as partial reasoning agents at the behavior level, our study proposes an approach for actually building partial reasoning agents, i.e. as opposed to Alver & Precup (2023), in our study we propose an interpretable architecture that allows for actually knowing that the agent is dynamically attending to the relevant aspects of the environment.

**Attention in RL.** Attention mechanisms have found many uses in RL (see e.g., Sorokin et al., 2015; Zambaldi et al., 2018; Mott et al., 2019; Tang et al., 2020; Liu et al., 2020; Tang & Ha, 2021). Different from these studies, which use attention in the context of model-free RL, our work here specifically uses attention to build partial reasoning agents. Specifically, we utilize attention for the purposes of aspect identification and (soft) filtration in high-dimensional domains.

**Value-Equivalent Planning.** A recent trend in model-based RL is to learn models that directly optimize for value-based planning, which has been studied under the name of value-equivalent planning (see e.g., Silver et al., 2017a; Farquhar et al., 2017; Oh et al., 2017; Schrittwieser et al., 2020; Grimm et al., 2020; 2021). Our work also advocates the idea that models should optimize for value-based planning and uses a value-equivalent simulation module. However, our work differs in that we are also explicit in semantics of the representations, i.e. we restrict them to be the different aspects of the environment, which allows for better representation learning. We also demonstrate that this explicitness can provide significant advantages in generalization.

**Cognitive Science.** Performing partial reasoning also has connections to the conscious planning literature in cognitive science (see e.g. Baars, 1993; 2002; Dehaene et al., 2021; Van Gulick, 2022; Bengio, 2017; Goyal & Bengio, 2022). In conscious planning, it is hypothesized that humans perform planning with a few abstract entities that are obtained by attending to the relevant aspects of the environment. This type of planning has been theorized to allow for efficient generalization (Bengio, 2017; Goyal & Bengio, 2022). Similar to Zhao et al. (2021), our work can also be viewed as a study that incorporates some of the ideas from this literature to the model-based RL literature.

**Interpretable RL.** Our work is also related to the line of research that aims for building interpretable RL agents (Jiang & Luo, 2019; Druce et al., 2021; Glanois et al., 2021; Di Langosco et al., 2022; Delfosse et al., 2023; 2024a;b; Bastani et al., 2018; Kohler et al., 2024; Luo et al., 2024). However, rather than leveraging interpretable architectures in safety-critical applications and for the main purpose of understanding the behavior of the agent, our study leverages them in a complementary fashion: for the purpose of shedding light on the internal working mechanisms of our partial reasoning agent and in understanding that it actually focuses on the relevant aspects of the environment.

## 6 Conclusion

To summarize, in this study, we have presented a novel approach for building partial reasoning agents which are agents that dynamically focus their reasoning on the relevant aspects of the environment. Unlike existing approaches, our approach works with pixel-based inputs and it allows for interpreting the focal points of the agent. Our empirical results (i) demonstrate that the proposed approach allows for effective generalization in high-dimensional domains with raw observational inputs, and (ii) show that it allows for building reasoning agents that indeed dynamically focus their reasoning on the relevant aspects of the environment throughout the agent-environment interaction. We believe that our approach can be an important step towards building scalable and interpretable reasoning agents that are able to effectively generalize to novel situations. Before closing, it is important to note that even though we have made use of particular attentive architectures in the modules, our overall architecture, presented in Fig. 1, is a general architecture and its modules can be implemented by using different attentive architectures. Finally, we note that in this study, similar to Zhao et al. (2021), we have only considered environments in which the details keep changing but the overall task remains the same. In future work, we hope to extend our work and tackle environments in which the overall task also changes over time.

### Acknowledgments

This project has been partly funded by an NSERC Discovery grant and the Canada CIFAR AI Chair program. We would like to thank the anonymous reviewers for providing critical and constructive feedback.

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

# A    Internal Details of the Modules

## A.1    Aspect Identification Module

As explained in the main text, the aspect identification module consists of (i) an encoder network, (ii) a slot attention module and (i) a decoder network. The architectural details of the encoder and decoder networks are summarized in Fig. 6 & 7, respectively.

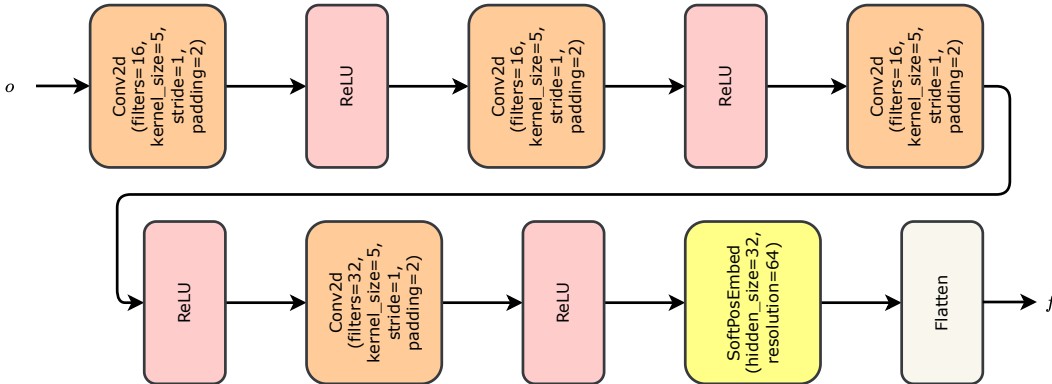

Figure 6: The architecture of the encoder network. The black arrows indicate the direction of information flow.

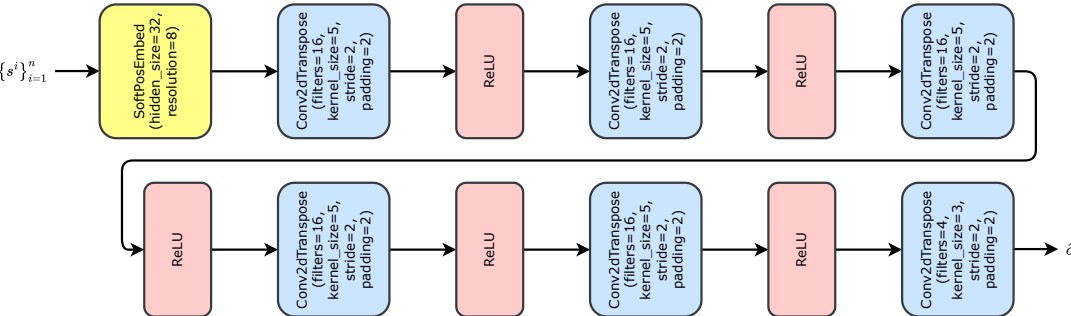

Figure 7: The architecture of the decoder network. The black arrows indicate the direction of information flow.

For the implementation of the slot attention module, we have followed the pseudocode in Locatello et al. (2020), presented in Algorithm 1, and have used the hyperparameters in Table 1. In our experiments, we have experimented with 4, 6, 8, 10, 16 slots and all of them resulted around the same performance (score).

Table 1: The hyperparameters of the slot attention module.

| Parameters | Values |
|---|---|
| Batch size | 256 |
| Resolution | 64 |
| Number of slots ($n$) | 8 |
| Number of iterations ($T$) | 3 |
| Warmup steps | 1e4 |

Finally, we note that we have used the same aspect identification module across the two domains (MiniGrid and Procgen) considered in this study.

---

**Algorithm 1:** The pseudocode of the slot attention module (Locatello et al., 2020). Note that we use the same variable names with Locatello et al. (2020).

---

**1 Input:** `inputs`, `slots` $\sim \mathcal{N}(\mu, \text{diag}(\sigma))$
**2 Layer Parameters:** $q, k, v$: linear mappings for obtaining the query, key and value vectors; `GRU`, `MLP`
**3 for** $i = 0, \ldots, T$ **do**
**4**    `slots_prev = slots`
**5**    `attention = softmax` $\left(q(\texttt{slots}) \cdot k(\texttt{inputs})^\top, \texttt{axis='slots'}\right)$
**6**    `updates = weightedsum (weights=attention, values=`$v$`(inputs))`
**7**    `slots = GRU (state=slots_prev, inputs=updates)`
**8**    `slots += MLP(slots)`
**9 end**
**10 return** `slots`

---

## A.2 Permutation-Invariant Filtration Module

Internally, the PI filtration module is a PI soft attention mechanism (Lee et al., 2019), in which the key and value matrices $K, V$ are obtained by linear transformations of the slots and the query matrix $Q$ is a linear transformation of fixed positional embeddings, see Algorithm 2.

---

**Algorithm 2:** The pseudocode of the PI soft attention mechanism.

---

**1 Input:** `inputs`, `pos_encodings`
**2 Layer Parameters:** $q, k, v$: linear mappings for obtaining the query, key and value vectors
**3** `score =` $q(\texttt{pos\_encodings}) \cdot k(\texttt{inputs})^\top$
**4** `att_weights = softmax (score, axis='inputs')`
**5** `output = weightedsum (weights=att_weights, values=` $v$`(inputs))`
**6 return** `output`

---

## A.3 Value-Equivalent Simulation Module

The value-equivalent simulation module consists of a dynamics and prediction network. The architectural details of these networks are summarized in Fig. 8 & 9. For the sake of presentation clarity we omit the time subscripts.

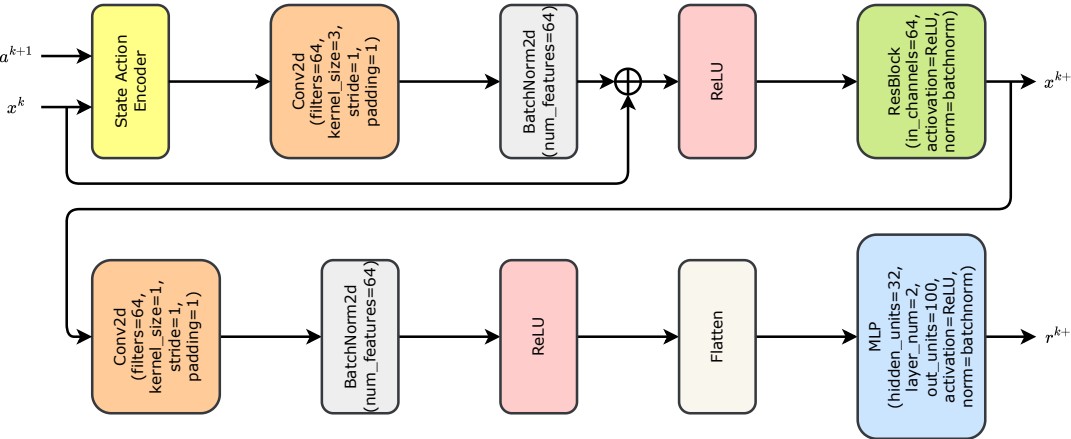

Figure 8: The architecture of the dynamics network. The black arrows indicate the direction of information flow.

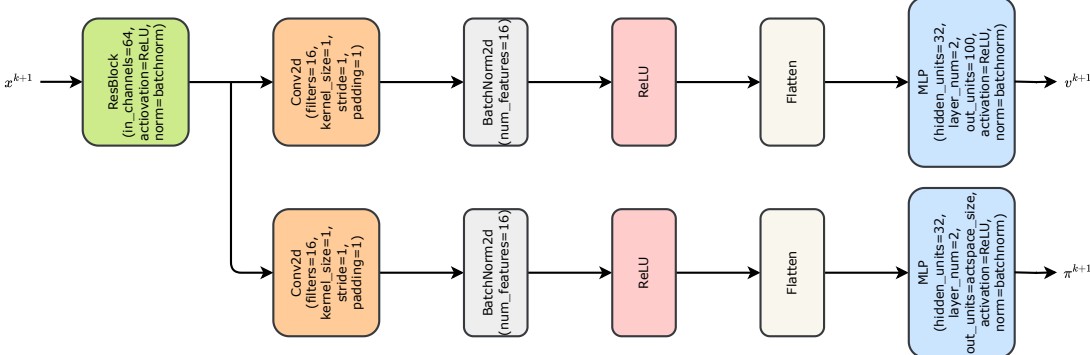

Figure 9: The architecture of the prediction network. The black arrows indicate the direction of information flow.

## B  Details of the Training Procedure

As explained in the main text, we train our proposed architecture (see Fig. 1), in an end-to-end manner, with the loss function $\mathcal{L}_{\text{total}}$ depicted in Eq. 1, which is a weighted sum of the reconstruction loss $\mathcal{L}_{\text{recon}}$ and the simulation loss $\mathcal{L}_{\text{sim}} = \mathcal{L}_p + \mathcal{L}_v + \mathcal{L}_r$. In our experiments, we ended up in using an equal weighting for the losses as the different combinations of $\alpha \in [0, 1]$ and $\beta \in [0, 1]$ did not lead to a significant increase or decrease in the generalization performances, i.e. we set $\alpha$ and $\beta$ to 0.5.

In $\mathcal{L}_{\text{total}}$, the reconstruction loss $\mathcal{L}_{\text{recon}}$ is simply the squared error between the input observations $o$ and the reconstructed observations $\hat{o}$:

$$\mathcal{L}_{\text{recon}}(o, \hat{o}) = (o - \hat{o})^2, \tag{2}$$

And, the policy, value function and reward losses are simply cross-entropy losses:

$$\mathcal{L}_p(\pi, p) = \boldsymbol{\pi}^\top \log \mathbf{p}, \tag{3}$$

$$\mathcal{L}_v(z, v) = \boldsymbol{\phi}(z)^\top \log \mathbf{v}, \tag{4}$$

$$\mathcal{L}_r(u, r) = \boldsymbol{\phi}(u)^\top \log \mathbf{r}, \tag{5}$$

where $\boldsymbol{\phi}(\cdot)$ refers to a transformation from a scalar representation to a categorical one, and the $\mathbf{p}$, $\mathbf{v}$ and $\mathbf{r}$ indicate the categorical outputs of the dynamics and prediction networks. Note that, in all of the loss functions, the loss is over batches of data and the time subscripts are omitted for simple presentation.

Finally, we also note that the total loss $\mathcal{L}_{\text{total}}$ in Eq. 1 also contains an weight decay term. However, for the sake of simplicity, we have omitted it in the presentation.

## C  Implementation Details of the Agents

**PR Agent.** The PR agent is a partial reasoning agent that was built by our proposed approach in Sec. 3. This agent is implemented by using the MCTS simulation framework of Niu et al. (2024).[3] More specifically, we have built upon the available MuZero implementation by replacing the internals of the `RepresentationNetwork` class in `lzero/model/common.py` with the internals of the aspect identification

---

[3]See `https://github.com/opendilab/LightZero` for the publicly available code.

and PI filtration modules (see App. A.1 & A.2). The hyperparameters of this agent are presented in Table 2.

Table 2: The hyperparameters of the PR agent.

| Parameters | Values |
|---|---|
| Weight of policy loss | 1 |
| Weight of value loss | 1 |
| Weight of reward loss | 1 |
| Number of MCTS simulations | 10 (MiniGrid)
25 (Procgen) |
| Reanalyze ratio | 0 |
| Number of frames stacked | 1 |
| Number of frames skip | 1 (MiniGrid)
4 (Procgen) |
| Length of game segment | 400 |
| Replay buffer size (in transitions) | 1e6 |
| TD steps | 5 |
| Number of unroll steps | 5 |
| Batch size | 256 |
| Model update ratio | 0.25 |
| Reward clipping | True |
| Optimizer type | Adam |
| Learning rate | 1e-4 |
| Discount factor | 0.99 |
| Frequency of target network update | 100 |
| Weight decay coefficient | 1e-4 |
| Max gradient norm | 10 |
| Discrete action encoding type | True |
| Priority exponent coefficient | 0.6 |
| Priority correction coefficient | 0.4 |
| Dirichlet noise weight | 0.25 |

**MZ Agent.** The MZ agent is a MuZero (Schrittwieser et al., 2020) agent. For this agent, we have used the already available `MuZero` implementation in Niu et al. (2024) which directly follows the implementation of Schrittwieser et al. (2020). The architecture of its representation network is depicted in Fig. 10. For a fair comparison, we use the same hyperparameters with the PR agent, i.e. the hyperparameters in Table 2.

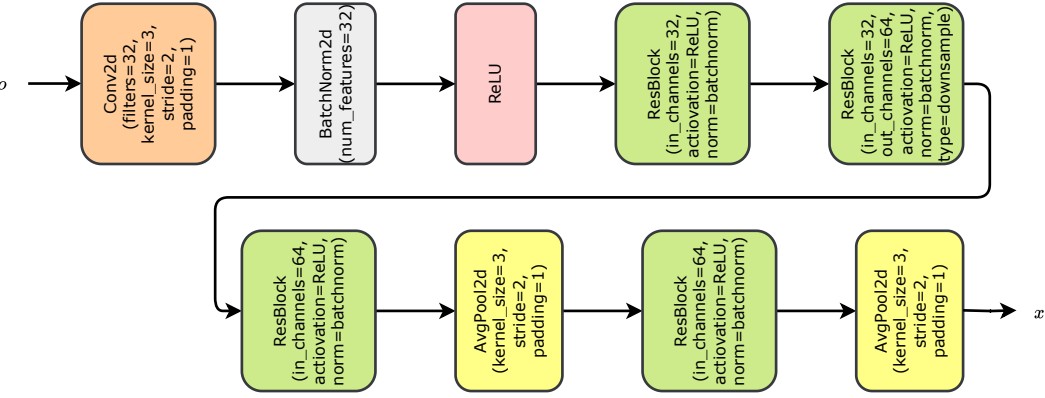

Figure 10: The architecture of the representation network of the MuZero (MZ) agent. The black arrows indicate the direction of information flow.

**MF Agent.** The MF agent is a model-free counterpart of the MZ agent, i.e. a version of the MZ agent in which no MCTS is performed: action selection is done solely with the value predictions from the root state. This agent shares the same architecture and hyperparameters with the MZ agent.

**PR-NoPIF Agent.** The PR-NoPIF agent is a version of the PR agent with no PI filtration module, i.e. a version in which the aspect identifying slots are directly passed to the value-equivalent simulation module. This agent shares the same hyperparameters with the PR agent.

**PR-SH Agent.** The PR-SH agent is a version of the PR agent which makes use of top-k semi-hard attention (rather than soft attention) in the implementation of its PI filtration module (see Algorithm 3). For this agent, we have experimented with different values of $k \in \mathbb{Z}^+$, ranging between 2 to 7, and ended up with setting it to 3 as it gave the best results (note that this also what was done in Zhao et al. (2021)). This agent shares the same hyperparameters with the PR agent.

---

**Algorithm 3:** The pseudocode of the PI top-k semi-hard attention mechanism (Ke et al., 2018; Zhao et al., 2021). The `topk_mask()` function keeps the top `k` scores and sets all the others to `-inf`.

---

**1** **Input:** `inputs`, `pos_encodings`, `k`
**2** **Layer Parameters:** $q, k, v$: linear mappings for obtaining the query, key and value vectors
**3** `score` $= q($`pos_encodings`$) \cdot k($`inputs`$)^\top$
**4** `masked_score` $=$ `renormalize(topk_mask(score, k=k))`
**5** `att_weights` $=$ `softmax (masked_score, axis='inputs')`
**6** `output` $=$ `weightedsum (weights=att_weights, values=`$v($`inputs`$))`
**7** **return** `output`

---

## D  Additional Experimental Results

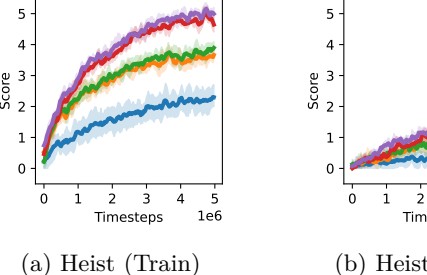

(a) Heist (Train)    (b) Heist (Test)

Figure 11: The generalization performances of the PR and baseline agents on the Heist environment. The plots are obtained again by evaluating the agent on a set of test levels (referred to as Test) as it gets trained on a set of training levels (referred to as Train) of the same environment. The training set consists of 500 randomly-sampled game levels. The test set consists of all of the possible game levels. The means and the confidence intervals are again computed over 15 independent runs.

To strengthen our quantitative and ablation results, we also compare the generalization performances of the PR and baseline agents on the Heist environment. In this environment, the agent (a stealer) must steal the gem that is behind a network of locks. Similar to every Procgen environment, a new game level is procedurally generated after every episode. For more details on the Heist environment, we refer the reader to the study of Cobbe et al. (2020). We see that the performance curves in Fig. 11 corroborate the findings of Sec. 4.3 & 4.4.

