# OpenReview forum: "An Attentive Approach for Building Partial Reasoning Agents from Pixels"
_TMLR — Accepted by TMLR_

### Review · Reviewer_pXif · 2024-07-03

**Summary Of Contributions:**

The authors propose an approach for building partial reasoning RL agents that attend to relevant parts of the high-dimensional input, e.g., pixels.

To achieve this, the authors propose three modules:
1. Aspect identification: to find the 'masks' that highlight the relevant inputs
2. Permutation-invariant filtration: filters the identified objects/regions for downstream decision-making
3. Value equivalent simulation: plans using the filtered objects as the initial state

**Audience:**

Yes

**Claims And Evidence:**

Yes

**Requested Changes:**

-Address the weakness (critical for recommendation)
-The code for reproducibility is missing (which would strengthen the work).
- Could you comment on how this scales to higher dimensional environments or different observational inputs?
Clarify the meaning of aspect, iteration, and time step (critical for recommendation) early in the manuscript

-

**Strengths And Weaknesses:**

Strengths:
- Outperforms baseline methods and ablations on the chosen example
- End-to-end trainable architecture
- The presented approach can directly use pixels as inputs

Weaknesses:
- Only a handful of examples are provided. Are there any failure cases? If yes, what are they? I encourage the authors to provide some additional examples in the appendix.
- Ablations on algorithm parameters are missing, e.g., what is the impact of the number of slots (on performance and the computational time)? Does this number have to be chosen according to the number of objects in the scene? What happens if there are fewer or more slots available?
- Why were the networks for the BigFish or CoinRun examples not trained until convergence?
-The environment variability is not fully specified, e.g., how many games are possible? Is it possible to transfer aspect identifiers, filters, or planners between environments?

---

### Review · Reviewer_dVqb · 2024-07-03

**Summary Of Contributions:**

Recent work has proposed partial reasoning models that aim to use a subset of the environment's objects in predictions. Such models have had the limitation of requiring information about the objects. The paper proposes an architecture for end-to-end training with partial reasoning models. Visualizations demonstrate that, during training, one component of the model (Aspect Identification Module) extracts objects from raw images and another component (Permutation-Invariant Filtration Module) subselects the objects in a soft manner to be used in prediction. Experiments are conducted in the generalization setting where the training and evaluation environments are slightly different. The results confirm that the algorithm with the proposed architecture outperforms a model-free or a simpler model-based algorithm and ablation studies show that partial reasoning is crucial to the performance.

**Audience:**

Yes

**Claims And Evidence:**

Yes

**Requested Changes:**

Clarifying the points in the previous section.

**Strengths And Weaknesses:**

The paper is clearly written and well organized. The experiments are well designed and the results are conclusive. Although it appears that both the Aspect Identification Module and Permutation-Invariant Filtration Module have been introduced in previous work for other purposes, introducing the current overall architecture along with stable training and evaluations and visualizations is important in my opinion. The final decision will depend on discussions with other reviewers. Meanwhile I ask the authors to clarify these points to help with understanding the algorithm:

1. What makes the Slot Attention Module bind to objects? Although the visualizations make it clear that this part of the Aspect Identification Module discovers slots that correspond to the objects in the environment, the current text in Section 3.1 does not describe why this behavior emerges from this structure. Have Locatello et al. provided an intuitive explanation? If so, bringing that explanation here is important.

2. Is the use of slot attention module for this purpose novel? As the submitted paper states, this module was introduced before by Locatello et al.. My question is whether this module was originally proposed to help with model learning and to extract objects in environments similar to the current experiments or was it originally proposed and evaluated in a completely different context. If the latter is the case, showcasing the efficacy of this module in these environments is an extra contribution of the submission. While in my opinion the submission is impactful enough either way, mentioning the original motivation for this module is helpful for situating the submission in the literature.

3. What is the target for the alpha mask? The decoder outputs 4 channels which are RGB and the alpha mask and is trained to reconstruct the input. Does this mean that the input has an alpha mask? If not, how is the alpha mask channel trained?

4. It appears that the architecture is based on vision transformer where the sequence is made of patches of pixels. If this is the case, it should be mentioned in the main text as currently it is not clear what the sequence is.

Minor comment: There are quite a few typos through the text that I can list in the later stages of the review process. The most important one is the definition of d_0 in the last line of page 2 as a function from the state space to distributions. The function d_0 is a itself a distribution, not a function to distributions.

---

### Review · Reviewer_rKkT · 2024-07-08

**Summary Of Contributions:**

The paper presents a novel architecture for model-based RL. Compared to other architectures, the proposed one uses an attention module that allows the agent to identify important components of the environment and to generalize to different tasks.
The authors validate their claim on a few environments.

EDIT:
The authors have addressed my concerns and are planning to add more experiments and release the code for the camera-ready version.

**Audience:**

Yes

**Broader Impact Concerns:**

No concern.

**Claims And Evidence:**

Yes

**Requested Changes:**

See "strenghts and weaknesses". To summarize:
- Discuss more about generalization
- More experiments (more MiniGrid or Procgen, or other environments like MiniWorld would be really nice)
- More baselines
- Better explanation of Figure 3
- Release code (or at least plan to)

**Strengths And Weaknesses:**

I am familiar with model-based RL, less with attention training. My understanding is that the authors combined existing architectures to achieve generalization, claiming that their agent is able to perform attention-based reasoning, focusing on important aspects of the environment.
The authors also claim that, unlike prior work, their architecture 1) works from pixels, 2) allows to interpret what the agent focuses on.

The paper is well written and easy to read, but I am not sure it has enough novelty and if experiments are sufficient for the paper to be accepted.

1) Experiments are very limited (only 2 pairs of environment/tasks), even though the authors write "we compare the generalization performances [...] across a **variety** of image-based MiniGrid and Procgen domains".
2) I am not sure how to interpret Figure 3. I would expect that the agent is trained on one environment (for example CoinRun) and then tested on another (e.g., BigFish). But the caption only say where the agent was trained OR tested. For example, in (a) the agent was trained on DynObs-Pass, but where was it tested?
3) I believe that the main contribution of the paper is in Sec 4.5, but it's very short and I would have expected more. This is probably due to the lack of overall experiments (few environments). Figure 4 and 5 are really interesting, but it would be much better to really have a variety of domains where the architecture is evaluated.
4) I think comparison with other approaches are also necessary. I am not very familiar with attention-based architecture, but how would this compare to classic VAE? Wouldn't a VAE be able to identify important pixels? In the end, the background is always black and it should be easy for a VAE to give importance to non-black pixels denoting objects and the agent.
5) I am a bit confused by the kind of "generalization" the authors aim for. It seems to be "task generalization" (even though not state explicitly, it is written that the agent needs to learn on relevant aspects of the environment, and then apply its knowledge to new task). However, "generalization" could also be "environment generalization" (same task, different environments). But isn't this what also happens in Procgen? CoinRun and BigFish are two different environment with two different tasks. Does it mean that the architecture can generalize across both tasks and environments?
6) While there is (limited) empirical evidence about why the PI module is important, this is never explained formally. Instead, the authors always refer to Zhao et al. (2021). It would be better (and make the paper easier to understand) if the authors would add a minimal explanation of why the PI module works.
6) There is no mention to ever releasing any code. This is an empirical paper and I believe it is important to provide code to replicate results.

A few more notes:
- Figure 1 should be large, the text within it is very hard to read.
- About MiniGrid, the authors write "otherwise the episode terminates without any reward". In MiniGrid, this is not true (unless the authors have changed the implementation). The agent always receive 0 rewards, which are still rewards. The sentence should be changed to "otherwise the episode terminates with only 0 rewards" (or something like that).

---

> ### Comment · Reviewer_rKkT · 2024-07-20
>
> Thank you for your response.
>
> * 1. I don't think that the fact that prior work compared on limited environment is a sufficient reason not to evaluate on more environment. Especially because this is a paper focused on empirical contributions. As you discussed in response 3.a, Sec 4.3, 4.4, 4.5 are (empirical) contributions. I am happy that you will include additional results for the camera-ready, though, and that should suffice.
>
> * 4. I am not very familiar with other approaches, but I agree with your explanation about why VAE would not work. I think it is work discussing it in the paper, since readers could have the same questions I had.
>
> * 6. "Limited" was referred to the few experiments. You have already promised more results for the camera-ready version.
>
> Nothing to say about the other points, you have address my concerns.
>
> The "No" about claims and evidence was motivated by the lack of experiments, the confusion from Figure 3, and not understanding well what was novel and what was not (a similar concern was raised by reviewer dVqb, to whom you properly replied). I have updated my review.

---

### Decision · Action_Editor_a1bf · 2024-08-07

**Recommendation:** Accept with minor revision

**Comment:**

The reviewers generally agree that, following the discussion period and revision of the manuscript, the paper meets the TMLR criteria for acceptance. The authors have stated that they intend to add experimental results in a new environment. Based on the discussion, I think this additional experiment should be added but the reviewers have already approved the experimental setup and I do not think the outcome of the experiment would make or break the paper, so my instinct is to treat this as a minor revision.

**Audience:**

The reviewers agree that the paper addresses an interesting problem. Noting that components of the proposed agent have been introduced in prior work, the reviewers generally agree that the paper presents and evaluates a novel combination/application of these ideas that may be of interest to some subsets of the TMLR audience.

**Claims And Evidence:**

The reviewers generally agree that the experiments are well-designed evaluations with appropriate baselines and ablations. Though the reviewers note that the small number of experimental domains is a limitation of the paper, the overall conclusion seems to be that the experiments provide sufficient empirical support for the central claims. The authors have mentioned that they intend to augment the paper with results in an additional Procgen environment, which will further strengthen the paper.